# Succinate induces skeletal muscle fiber remodeling via SUNCR1 signaling

Tao Wang[1,†], Ya-Qiong Xu[1,†], Ye-Xian Yuan[1,†], Ping-Wen Xu[2], Cha Zhang[1], Fan Li[1], Li-Na Wang[1], Cong Yin[1], Lin Zhang[1], Xing-Cai Cai[1], Can-Jun Zhu[1], Jing-Ren Xu[1], Bing-Qing Liang[1], Sarah Schaul[2], Pei-Pei Xie[1], Dong Yue[1], Zheng-Rui Liao[1], Lu-Lu Yu[1], Lv Luo[1], Gan Zhou[1], Jin-Ping Yang[1], Zhi-Hui He[1], Man Du[1], Yu-Ping Zhou[1], Bai-Chuan Deng[1], Song-Bo Wang[1], Ping Gao[1], Xiao-Tong Zhu[1], Qian-Yun Xi[1], Yong-Liang Zhang [1], Gang Shu[1,3,*] (ID) & Qing-Yan Jiang[1,3]

## Abstract

The conversion of skeletal muscle fiber from fast twitch to slow-twitch is important for sustained and tonic contractile events, maintenance of energy homeostasis, and the alleviation of fatigue. Skeletal muscle remodeling is effectively induced by endurance or aerobic exercise, which also generates several tricarboxylic acid (TCA) cycle intermediates, including succinate. However, whether succinate regulates muscle fiber-type transitions remains unclear. Here, we found that dietary succinate supplementation increased endurance exercise ability, myosin heavy chain I expression, aerobic enzyme activity, oxygen consumption, and mitochondrial biogenesis in mouse skeletal muscle. By contrast, succinate decreased lactate dehydrogenase activity, lactate production, and myosin heavy chain IIb expression. Further, by using pharmacological or genetic loss-of-function models generated by phospholipase Cβ antagonists, SUNCR1 global knockout, or SUNCR1 gastrocnemius-specific knockdown, we found that the effects of succinate on skeletal muscle fiber-type remodeling are mediated by SUNCR1 and its downstream calcium/NFAT signaling pathway. In summary, our results demonstrate succinate induces transition of skeletal muscle fiber via SUNCR1 signaling pathway. These findings suggest the potential beneficial use of succinate-based compounds in both athletic and sedentary populations.

**Keywords** aerobic exercise; fiber type; skeletal muscle; succinate; SUNCR1
**Subject Categories** Metabolism; Musculoskeletal System

## Introduction

In mammals, skeletal muscle comprises about 55% of the individual body mass [1,2]. Skeletal muscle is heterogeneous and composed of slow- and fast-twitch fiber types, which differ in contractile-protein composition, oxidative capacity, and substrate preference for ATP production [3]. Slow-twitch fibers have more myoglobin, more mitochondria [4], a higher level of intracellular calcium concentrations [5], and higher activity of oxidative metabolic enzymes than fast-twitch fibers. Therefore, the switching of skeletal muscle fiber from fast twitch to slow twitch is important for sustained and tonic contractile events [6,7], maintenance of energy homeostasis [8], and alleviation of fatigue.

Endurance or aerobic exercise is crucial to muscle fiber-type remodeling by increasing the mechanical and metabolic demand on skeletal muscle [9]. Previous study showed endurance training increases intracellular calcium concentration ($[Ca^{2+}]_i$) [10,11], which activates the calcineurin/nuclear factor of activated T cells (NFAT) [12,13] and myocyte enhancer factor-2 (MEF2) [14]. These two transcription factors play a dominant role in muscle formation and fiber remodeling. In addition to transient elevation of $[Ca^{2+}]_i$, endurance exercise also increases several specific TCA cycle intermediates, among which succinate increases the most [15,16]. However, whether these intermediates mediate endurance exercise-induced muscle fiber transition is rarely investigated. Succinate has been reported to induce cardiomyocyte hypertrophy [17] and osteoclastogenesis [18]. It also plays an important role in energy [19] and glucose [20] homeostasis by regulating mitochondrial oxygen consumption [21] and heat production from brown adipose tissue (BAT) [22]. Therefore, we hypothesize that succinate regulates skeletal muscle fiber remodeling.

To test this hypothesis, we first examined the effects of succinate on skeletal muscle fiber composition, metabolism, and exercise tolerance. By combining pharmacological and siRNA-mediated

1 Guangdong Province Key Laboratory of Animal Nutritional Regulation, College of Animal Science, South China Agricultural University, Guangzhou, Guangdong, China
2 Division of Endocrinology, Department of Medicine, The University of Illinois at Chicago, Chicago, IL, USA
3 National Engineering Research Center for Breeding Swine Industry, College of Animal Science, South China Agricultural University, Guangzhou, Guangdong, China
*Corresponding author. Tel: 86 020 85284901; E-mail: shugang@scau.edu.cn
†These authors contributed equally to this work as first authors
[Correction added on 5 September 2019, after first online publication: the article title has been corrected.]

knockdown model both *in vitro* and *in vivo*, we demonstrated that succinate induces skeletal muscle transition from fast twitch to slow twitch through the SUNCR1 signaling pathway. Our results indicate potential use of succinate as a dietary supplement to improve physical fitness and counteract fatigue.

# Results

### The dietary supplement of succinate shifts skeletal muscle fiber size distribution

To determine the effects of succinate on skeletal muscle growth, we fed male C57BL/6J mice with chow diet supplemented with 0, 0.5%, or 1% succinic acid disodium salt for 8 weeks. We found that succinate-supplemented diet increased serum SUA level (Fig 1A) but had no effects on the body weight gain (Fig 1B), food intake (Fig EV1A), fat mass (Fig 1C), lean mass (Fig 1D), gastrocnemius muscle index (Fig 1E), or liver index (Fig EV1B). Additionally, consistent with our previous report [23], we found that succinate activated Akt/mTOR cascade and inhibited FoxO3a (Fig EV1C and D). Interestingly, we also found that 1% succinate increased the

proportion of small muscle fiber (200–400 $\mu m^2$), while decreased the proportion of large muscle fiber (600–800 $\mu m^2$; Fig 1F and G). This shift of muscle fiber size distribution indicates that succinate may affect skeletal muscle contraction properties.

### Succinate enhances endurance exercise capacity and reduces muscle fatigue

To further investigate the effects of succinate on skeletal muscle contraction properties, we first tested the exercise capacity of mice. We found that succinate dose-dependently increased muscle grip strength (Fig 2A), low-speed running time (Fig 2B), and decreased falling time in four-limb handing test (Fig 2C). However, high-speed running time was unchanged by succinate supplementation (Fig 2D), which indicates succinate may specifically improve endurance exercise performance, but not explosive exercise performance.

It is well-known that endurance exercise performance is determined by oxygen supply and muscle fiber type [19]. We first tested if the oxygen-carrying capacity of muscle was enhanced by succinate. We found although succinate slightly increased the number of red blood cells (RBC; Fig 2E) and the hemoglobin (HGB) level (Fig 2F), the extent of these increases is not comparable to the

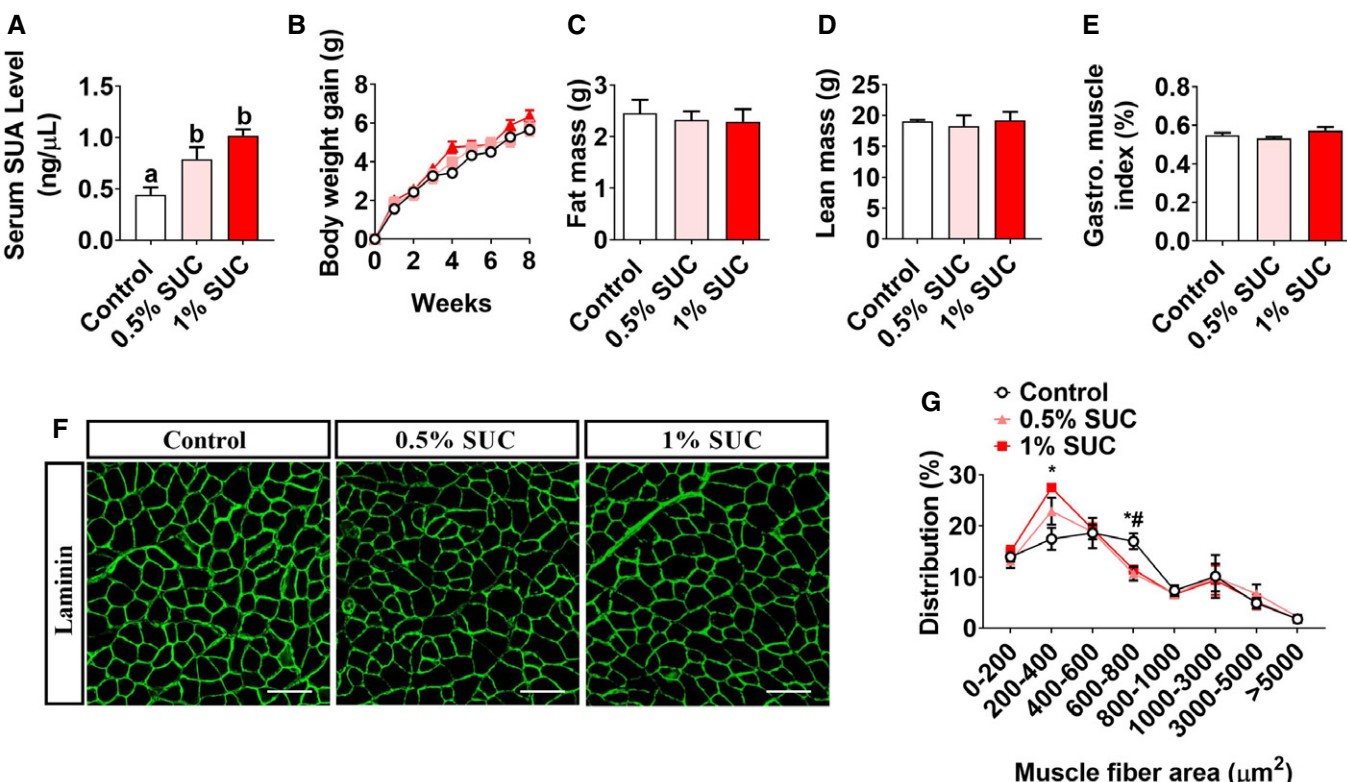

**Figure 1.  Effects of succinate on growth performance and serum concentration in mice.**

Male C57BL/6J mice were fed with chow diet supplemented with 0, 0.5, and 1% SUC for 8 weeks.

A–E   (A) Serum SUA level, (B) body weight gain, (C) fat and (D) lean mass and (E) gastrocnemius index.

F, G   (F) Gastrointestinal muscle fiber immunofluorescent laminin staining and (G) frequency histogram of fiber cross-sectional area. Scale bar in (F) represents 100 $\mu m$.

Data information: Results are presented as mean ± SEM (n = 6–8). Different letters between bars mean $P \leq 0.05$ in one-way ANOVA analyses followed by *post hoc* Tukey's tests. *: significant difference ($P \leq 0.05$) between 0.5% SUC and control group by non-paired Student's *t*-test. #: significant difference ($P \leq 0.05$) between 1% SUC and control group by non-paired Student's *t*-test.

                                                      

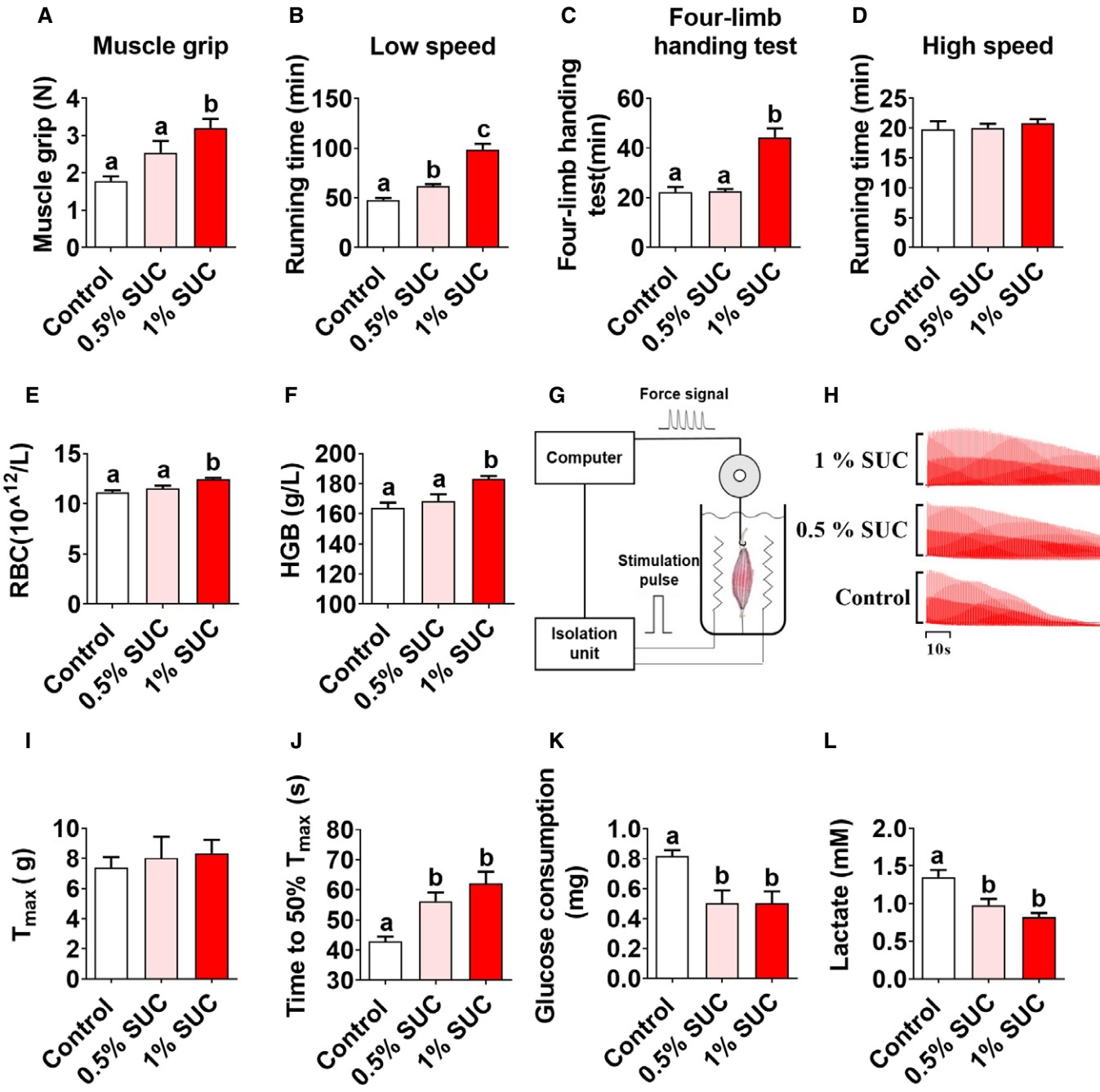

**Figure 2. Succinate enhances the endurance exercise capacity of skeletal muscle in mice.**

Male C57BL/6J mice fed with chow diet supplemented with 0, 0.5, and 1% SUC for 8 weeks.

A–D   (A) The muscle grip strength, (B) running time in low speed, (C) four-limb handing time, and (D) running time in high speed.

E, F   (E) Serum concentration of RBC and (F) HGB in whole blood.

G–L   (G–I) *Ex vivo* gastrocnemius muscle force, (J) fatigability, (K) glucose consumption, and (L) lactate production were tested.

Data information: Results are presented as mean ± SEM (*n* = 5–8). Different letters between bars mean *P* ≤ 0.05 in one-way ANOVA analyses followed by *post hoc* Tukey's tests.

dramatic improvement of endurance exercise capacity. In order to further characterize other parameters related to endurance exercise capability, we used an *ex vivo* strategy to evaluate isolated muscle contraction properties (Fig 2G). We found that dietary supplementation of succinate did not affect the maximum contractile force (Fig 2I), but significantly improved fatigue resistance of muscle (Fig 2H and J), with less glucose consumption (Fig 2K), and lactate production (Fig 2L) during contraction. Taken together, our

data indicate that succinate can increase oxygen-carrying capacity and reduce muscle fatigue.

## Succinate induces skeletal muscle fiber-type transition *in vivo*

There are four types of skeletal muscle fiber, including I, IIa, IIx, and IIb. Each of them expresses different myosin heavy chain and troponin isoforms. Here, we studied the effects of succinate on muscle fiber-type transaction in three different muscles, including soleus, extensor digitorum longus (EDL), and gastrocnemius. Soleus is known as a typical slow-twitch muscle (slow/slow), whereas EDL is a typical fast-twitch muscle (fast/fast). Gastrocnemius usually has

a lot of fast-twitch muscle fibers, or an equal number of fast and slow-twitch fibers (fast/slow mixed).

In mixed gastrocnemius muscle, we found that succinate upregulated slow-twitch fiber-associated genes MyHC I, MyHC IIa, PGC-1α, myoglobin, and TnnT1, whereas it downregulated fast-twitch fiber-associated genes, including MyHC IIb and TnnT3 (Fig 3A). Further, both Western blot (Fig 3B) and immunofluorescence (Fig 3C and D) demonstrated that succinate increased MyHC I/IIa protein expression and slow-twitch fiber percentage, while decreased MyHC IIb protein and fast-twitch fiber percentage. These results indicate that succinate induces a fast twitch to slow-twitch transition in skeletal muscle.

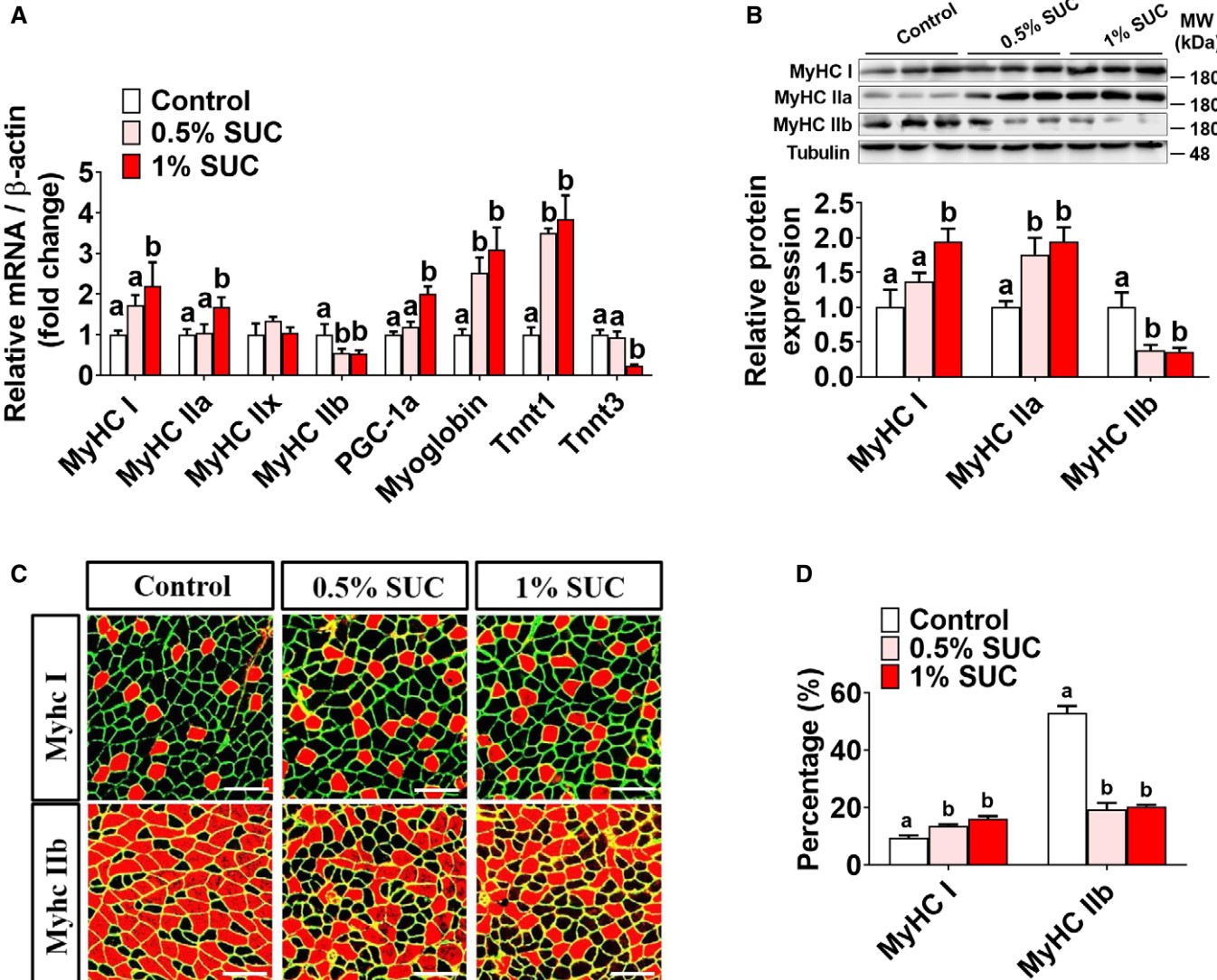

**Figure 3. Effects of succinate on MyHC expression in mice.**

Male C57BL/6J mice were fed with chow diet supplemented with 0, 0.5, and 1% SUC for 8 weeks.

A    The mRNA expression of MyHC I, MyHC IIa, PGC-1α, myoglobin, TnnT1 MyHC IIb, MyHC IIx, and TnnT3 in the gastrocnemius muscle (*n* = 5–6).

B    Immunoblots and quantification of MyHC I, MyHC IIa, and MyHC IIb protein expression in gastrocnemius (*n* = 3–4).

C, D    Representative images and quantification of laminin (green), MyHC I, and MyHC IIb immunofluorescent staining (red) in gastrocnemius (*n* = 3). Scale bar in (C) represents 100 μm.

Data information: Results are presented as mean ± SEM. Different letters between bars mean *P* ≤ 0.05 in one-way ANOVA analyses followed by *post hoc* Tukey's tests.

Consistently, we found that succinate dose-dependently increased MyHC I but not MyHC IIb protein expression in soleus, suggesting an increased proportion of slow-twitch fiber (Fig EV1E and F). On the other hand, succinate failed to affect the muscle fiber composition of EDL muscle (Fig EV1G and H).

Oxidative capacity of three muscles was also evaluated by the staining of succinate dehydrogenase (SDH), a marker of oxidative capacity of skeletal muscle at the fiber level. We found that succinate dose-dependently increased the percentage of SDH-positive fibers in SOL, EDL, and gastrocnemius muscles (Fig EV1I–N), suggesting succinate is

sufficient to improve mitochondrial content and oxidative capacity of mixed (gastrocnemius), slow/slow (SOL), or fast/fast (EDL) muscles.

### Succinate increases aerobic oxidation and mitochondrial biogenesis in skeletal muscle

A high number of mitochondrial and metabolic adaptation are generally accompanied with endurance exercise and skeletal muscle type transition [24]. Here, we tested the effects of succinate on metabolism and mitochondrial properties. We found that succinate increased

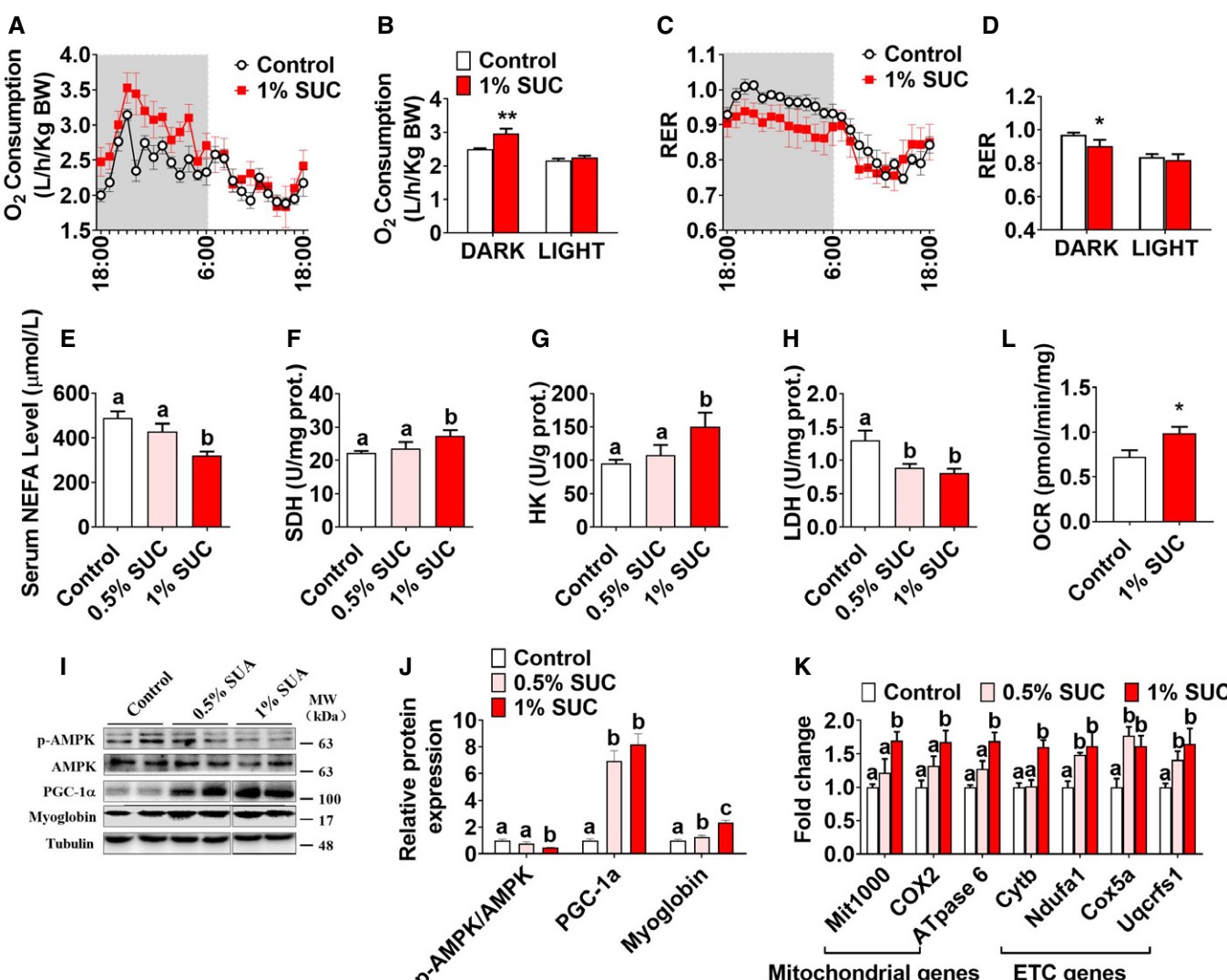

**Figure 4. Succinate promotes skeletal muscle mitochondrial biosynthesis and aerobic oxidation in mice.**

Male C57BL/6J mice were fed with chow diet supplemented with 0 and 1% SUC for 6 weeks.

A–D   The $O_2$ consumption ($VO_2$) (A, B) and respiratory exchange ratio (RER) (C, D).
E–H   Serum concentration of (E) NEFA in whole blood. The enzymes activity of (F) SDH, (G) HK, and (H) LDH in gastrocnemius.
I, J   Immunoblots and quantification of p-AMPK, PGC-1α, and myoglobin in gastrocnemius. The same lysates were used for the detection of PGC1α (100 kDa, Fig 4I), myoglobin (17 kDa, Fig 4I), myosin heavy chain (180 kDa, Fig 3B), and tubulin (48 kDa, shared in both Figs 3B and 4I).
K   Quantification of mitochondrial and electron transport chain (ETC)-related gene expression i respiratory exchange n gastrocnemius.
L   OCRs were measured under basal condition in gastrocnemius.

Data information: Results are presented as mean ± SEM (*n* = 4–6). Different letters between bars mean $P \leq 0.05$ in one-way ANOVA analyses followed by *post hoc* Tukey's tests. **P* ≤ 0.05 and ***P* ≤ 0.01 by non-paired Student's *t*-test.

whole-body oxygen consumption (Fig 4A and B) and decreased whole-body respiratory exchange ratio (RER; Fig 4C and D) in the dark cycle. In addition, serum non-essential fatty acid (NEFA) content was decreased by succinate supplementation (Fig 4E), suggesting that the decreased respiratory quotient may be attributed to the elevated fatty acid oxidation. Consistently, succinate enhanced the activity of succinate dehydrogenase (SDH; Fig 4F) and hexokinase (HK; Fig 4G) but suppressed the activity of lactic dehydrogenase (LDH; Fig 4H). These results suggest that succinate promotes aerobic metabolism. In supporting this point of view, an enhanced mitochondrial biogenesis was consistently shown in our model. When detecting the myosin heavy chain by WB, we also checked PGC1α and myoglobin protein simultaneously (Fig 4I and J), as well as the expression of genes related to mitochondria and electron transport chain (Fig 4K). These protein and mRNA expression level were dose-dependently increased by succinate in the gastrocnemius. However, the p-AMPK levels were reduced by succinate (Fig 4I and J), indicating that cellular energy status may not be the main reason for skeletal muscle type transition.

Based on a recent study showing that succinate increased adipose tissue metabolism and induced browning in high-fat diet (HFD)-induced obesity mice [22], we postulated that succinate has a similar stimulatory effect on metabolism in muscle. To test this, we further evaluated oxygen consumption in skeletal muscle and consistently found that succinate significantly increased oxygen consumption ratio (OCR) in the gastrocnemius (Fig 4L). Together, these results indicate that succinate induces skeletal muscle fiber remodeling by promoting mitochondrial biosynthesis and aerobic oxidation.

## Succinate induces fiber-type remodeling and increases mitochondrial content in C2C12 myotubes

To test the direct effect of succinate on skeletal muscle, we used C2C12 myotubes as an *in vitro* model to study the role of succinate in skeletal muscle fiber-type remodeling. Similar to the previous *in vivo* study, we found that succinate significantly increased the proteins and genes of slow-twitch fiber markers, while decreased the proteins and genes of fast-twitch markers as indicated by both immunofluorescence (Fig 5A and B) and qPCR (Fig EV2A). Regarding metabolic enzymes, succinate enhanced the activity of SDH (Fig 5C), but reduced the activity of LDH (Fig 5D) and lactic acid production (Fig 5E) in C2C12 myotubes.

Additionally, we tested the number, morphology, and activity of mitochondria. Consistent with our *in vivo* data, succinate significantly increased mitochondrial DNA content (Fig 5F), cellular mitochondrial density (Fig 5G–J), and coverage (Fig 5I). However, the size (Fig 5J) and the membrane potential of mitochondria (Fig EV2B and C) were not affected by succinate. These results suggest that the enhanced aerobic oxidation is mainly due to the increased mitochondrial number, but not the activity of each mitochondrion. These *in vitro* data reveal a direct role of succinate in the slow-twitch transition, mitochondrial biogenesis, and aerobic oxidation.

## SUNCR1/PLCβ/Calcium signaling pathway mediates succinate-induced fiber-type transition

To explore the intracellular mechanism for succinate-induced fiber-type transition, we compared the expression of SUNCR1, an endogenous receptor of succinate [25], in the soleus and gastrocnemius muscles. Interestingly, the protein (Fig 6A and B) and mRNA (Fig 6C) of SUNCR1 in the soleus (typical slow/slow muscle) are much higher than levels in gastrocnemius (typical mixed slow/fast muscle). In addition, exercise significantly increased SUNCR1 protein expression in both soleus and gastrocnemius muscles (Fig 6A and B), suggesting a potential role of this receptor in skeletal muscle fiber-type remodeling.

To test this point of view, we generated pharmacological or genetic loss-of-function models to investigate the requirement of SUNCR1 in succinate-induced muscle fiber-type transition. We found that succinate triggered a transient elevation of $[Ca^{2+}]i$ in C2C12 myotubes (Fig 6D) and promoted nucleic NFAT accumulation in the gastrocnemius muscle shortly (0.5–3 h) after acute succinate administration (Fig 6E–G). Importantly, pharmacological blockage of PLC-β, a key mediator of a GPCR-triggered calcium signaling pathway, effectively abolished succinate-induced $[Ca^{2+}]i$ elevation (Fig EV3A) and fiber-type transition (Fig EV3B–D) in C2C12 myotubes. Consistently, siRNA-mediated knockdown of SUNCR1 in C2C12 myotubes (Fig 6H) effectively abolished succinate-induced $[Ca^{2+}]i$ elevation (Fig 6I); myotube fiber conversion (Fig 6M–O); activity changes in SDH, HK, and LDH (Fig 6J–L); and lactate production (Fig EV3E). These *in vitro* data suggest that succinate-induced C2C12 myotube fiber switch is mediated by SUNCR1.

## SUNCR1 is required for succinate-induced skeletal muscle fiber switch *in vivo*

To determine the role of SUNCR1 in succinate-induced skeletal muscle fiber switching *in vivo*, we constructed a congenital SUNCR1 global knockout mouse model (Fig EV4A–C). We found that SUNCR1 null mice showed the same body weight gain (Fig EV4G), food intake (Fig EV4D), and body composition (Fig EV4E and F) as their wild-type littermates. Additionally, we found the stimulatory effects of succinate on AKT/mTOR/FOXo3a pathway were diminished in SUNCR1 KO mice (Fig EV4H and I), suggesting a SUNCR1-mediated activation on protein synthesis. Interestingly, SUNCR1 KO also effectively blocked the regulatory effects of succinate on oxygen consumption (Fig 7A and B), RER (Fig 7C and D), and exercise capacities, including slow-speed running time (Fig 7G), four-limb handing time (Fig 7F), and muscle grip (Fig 7E). Consistently, the activities of SDH, HK, and LDH (Fig 7H–J); skeletal muscle fiber type (Fig 7K–N); and the expression of NFAT and PGC-1α (Fig 7K and L) failed to be changed by succinate in the gastrocnemius of SUNCR1 KO mice. These data support an essential role of SUNCR1 in succinate-induced skeletal muscle fiber switching.

Since SUNCR1 is universally expressed in most metabolic tissues, including adipose tissue, liver, and heart, succinate may indirectly act on SUNCR1 expressed in other metabolic tissues to regulate skeletal muscle metabolism and fiber switching. To exclude this possibility, we further generated and validated a gastrocnemius-specific SUNCR1 knockdown mouse model by gastrocnemius-specific injection of SUNCR1 siRNA lentivirus during adulthood (Fig 8A and B). Consistent with our observation in congenital SUNCR1 global knockout mice, SUNCR1 selective knockdown in the gastrocnemius muscle showed no effects on food intake (Fig EV5A), body weight (Fig EV5B), and body composition (Fig EV5C and D).

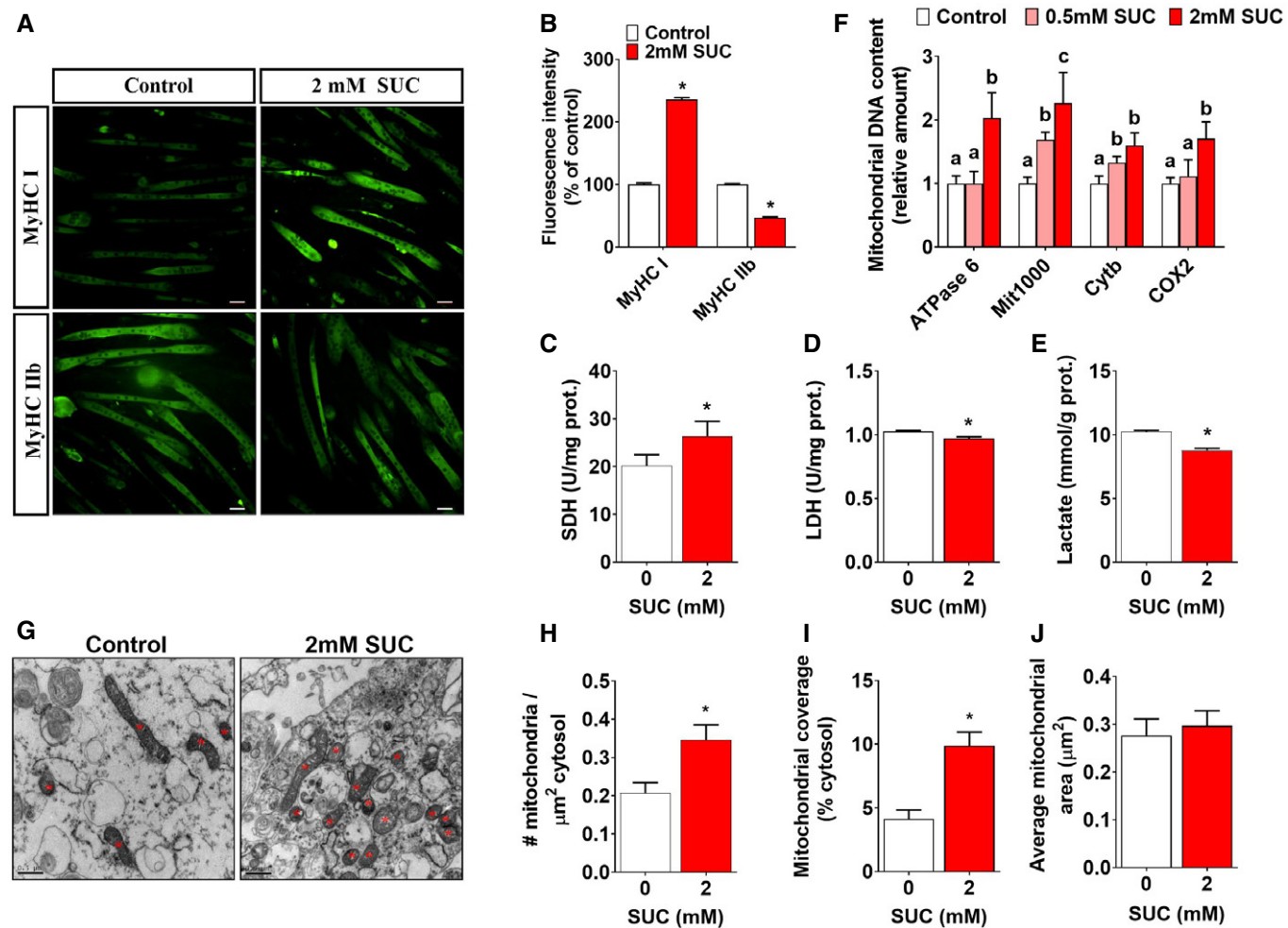

**Figure 5.  Effects of succinate on MyHC expression, mitochondria biosynthesis, and metabolism in C2C12 cells.**

C2C12 cells were treated with 0, 0.5, and 2 mM SUC for 48 h.

A, B   Representative images and quantification of MyHC I and MyHC IIb immunofluorescent staining (green) in C2C12 cells (*n* = 16).
C–E   The enzymes activity of (C) SDH, (D) LDH, and (E) lactate production in C2C12 cells.
F     Quantification of mitochondrial DNA contents in C2C12 cells.
G–J   (G) Mitochondrial electron microscopy showed the (H) mitochondrial density, (I) mitochondrial coverage, and (J) average mitochondrial area in C2C12 cell. Scale bar in (A) represents 50 µm; scale bar in (G) represents 0.5 µm.

Data information: Results are presented as mean ± SEM (*n* = 6–8). Different letters between bars mean $P \leq 0.05$ in one-way ANOVA analyses followed by *post hoc* Tukey's tests. *$P \leq 0.05$ by non-paired Student's *t*-test.

Importantly, SUNCR1 gastrocnemius-specific knockdown consistently attenuated the regulatory effect of succinate on exercise capacity (Fig 8C–E), muscle fiber type (Fig 8I and J), and related associated enzyme activity (Fig 8F–H). Together, these data support an indispensable role of muscle SUNCR1 in succinate-induced skeletal muscle fiber remodeling.

## Discussion

Skeletal muscle fiber types are distinguished by myosin heavy chain (MyHC) isoforms [26], metabolic enzyme activity [6], mitochondrial number [27], and contractile properties [28]. Endurance or aerobic exercise is well known as an effective way to induce skeletal muscle remodeling by increasing mechanical and metabolic demand on skeletal muscle [29–31]. Interestingly, exercise also dramatically elevates the content of several TCA cycle intermediates, including succinate [32]. Succinate previously has been shown to regulate mitochondrial function and reactive oxygen species production in muscle [33], which is a distinguishing feature of skeletal muscle fiber types [34]. Based on these observations, we speculate that succinate is a key mediator for exercise-induced muscle fiber remodeling.

In supporting this point of view, we found that dietary succinate supplementation improved the endurance exercise performance and attenuated skeletal muscle fatigability, accompanied by enhanced aerobic metabolism and upregulated MyHC I/IIa expression. These data demonstrated for the first time that succinate induces a switch from fast twitch to slow-twitch fibers,

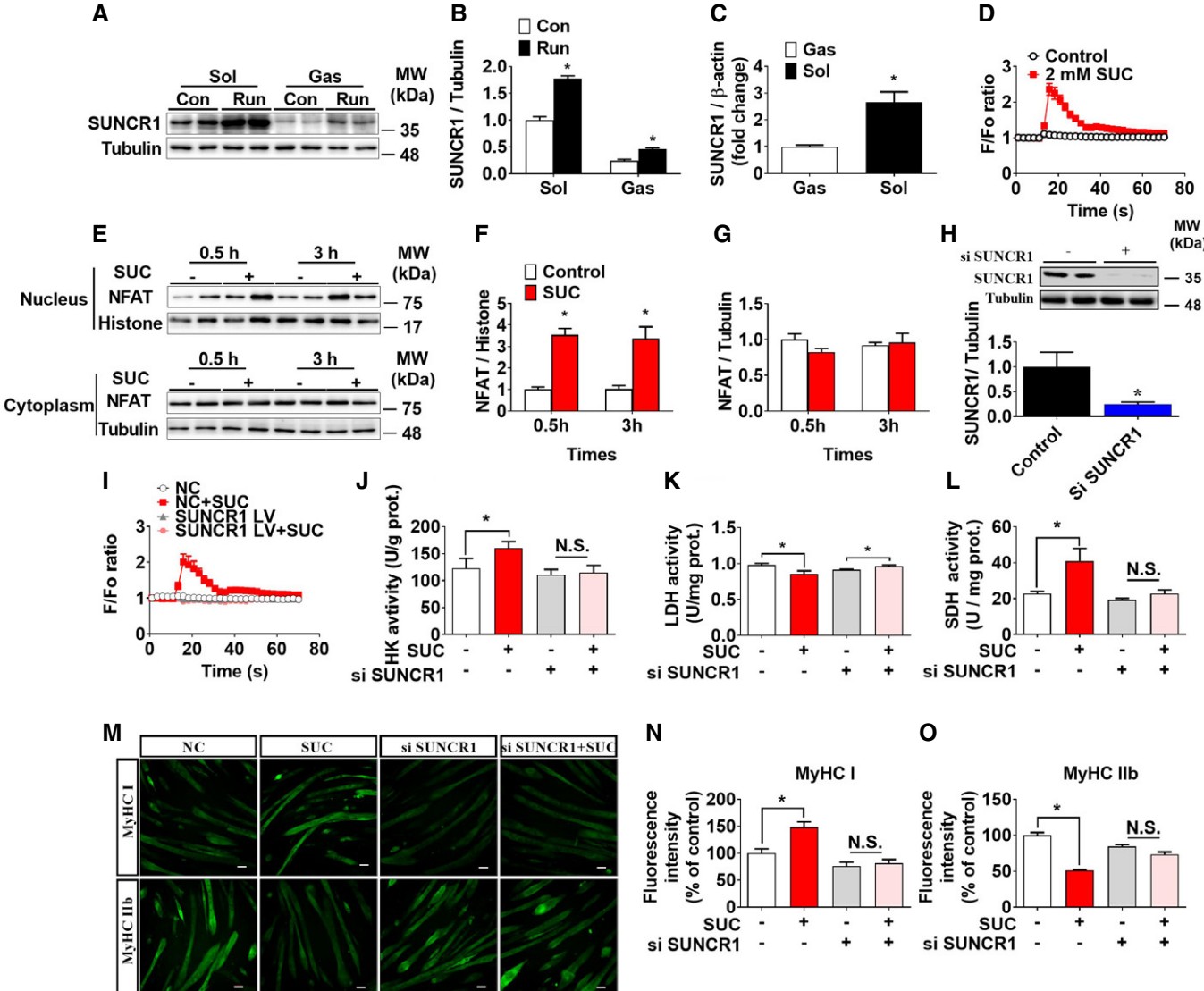

**Figure 6. SUNCR1 is required for succinate to induce the fiber-type transition in myotubes.**

A, B    SUNCR1 protein expression in the gastrocnemius from sedentary or post-running mice (*n* = 4).

C    The mRNA level of SUNCR1 in gastrocnemius and soleus (*n* = 7–8).

D    [Ca²⁺]i in C2C12 cells treated with 0 or 2 mM SUC (*n* = 18–20).

E–G    NFAT protein expression in nucleus and cytoplasm of gastrocnemius 0.5 h or 3 h after i. p. injection of 15 mg/kg succinate in C57BL/6J mice (*n* = 4).

H    SUNCR1 protein expression in C2C12 cells transfected with vector or siSUNCR1 (*n* = 3).

I–L    (I) [Ca²⁺]i, and enzymes activity (*n* = 9–10)of (J) HK, (K) LDH, and (L) SDH in vector or siSUNCR1 transfected C2C12 cells treated with 0 or 2 mM SUC (*n* = 5–6).

M–O    Representative images and quantification of MyHC I and MyHC IIb immunofluorescent staining (green) in C2C12 cells (*n* = 3). Scale bar in (M) represents 50 μm.

Data information: Results are presented as mean ± SEM. *$P \leq 0.05$ by non-paired Student's *t*-test.

suggesting a potential mechanism for metabolite-mediated skeletal muscle fiber-type transition.

Mitochondria are the main sites of cellular aerobic respiration. In general, cellular or tissue oxidative metabolism is enhanced by increasing the number of mitochondria [35]. PGC-1α has been shown to be a key regulator of mitochondrial biosynthesis and oxidative metabolic enzyme [36]. Overexpression of PGC-1α increases mitochondrial content and the oxidase levels of skeletal muscle, which results in more resistance to fatigue [37]. In this

study, we found that succinate increased the protein expression of PGC-1α, as well as the mitochondrial content both *in vitro* and *in vivo*. In addition, succinate further enhanced O₂ uptake in skeletal muscle cells. This observation is consistent with a previous study showing that succinate increases mitochondrial oxygen consumption in *ex vivo* skeletal muscle obtained from septic animals.

Besides the number of mitochondria, the function of mitochondria was also strengthened by mitochondrial membrane potential and mitochondrial membrane enlargement [38]. Thus, we further

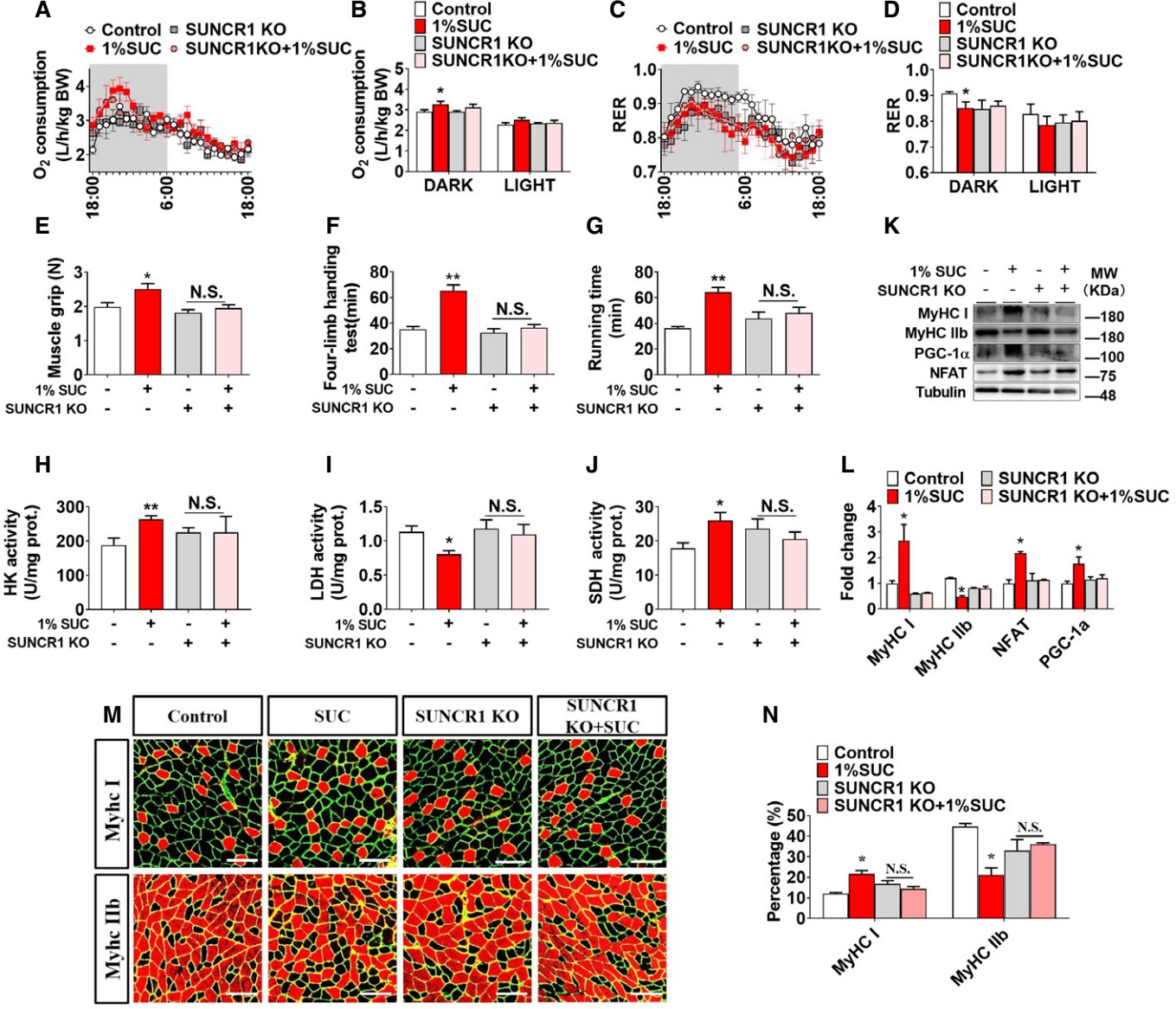

**Figure 7. SUNCR1 global knockout blocks the effect of succinate on muscle fiber switch *in vivo*.**

Male C57BL/6J or SUNCR1 KO mice were fed with chow diet supplemented with 0 or 1% SUC for 6 weeks.

A–G  (A, B) The O₂ consumption (VO₂), (C, D) RER, (E) muscle grip strength, (F) four-limb handing time, and (G) low-speed running time.

H–J  The enzymes activity of (H) HK, (I) LDH, and (J) SDH in gastrocnemius.

K, L  Immunoblots and quantification of MyHC I, MyHC IIb, NFAT, and PGC-1α protein in gastrocnemius.

M, N  Representative images and quantification of laminin (green), or MyHC I and MyHC IIb (red) immunofluorescent staining in gastrocnemius muscle (*n* = 3). Scale bar in (M) represents 100 μm.

Data information: Results are presented as mean ± SEM (*n* = 5–6). *$P \leq 0.05$ and **$P \leq 0.01$ by non-paired Student's *t*-test.

examined the morphology changes in mitochondria by succinate using an electron microscope. We found that succinate increased mitochondrial number without changing mitochondrial size. Although we were unable to examine the function of all signal mitochondria, enzyme activities and O₂ uptake strongly suggested that the increase in mitochondrial number is accounted for the enhanced mitochondrial function in skeletal muscles.

Skeletal muscle fiber-type remodeling involves several key signaling pathways, including calcium [39] and AMPK [40]. In this study, we found that succinate boosted $[Ca^{2+}]_i$ and increased the protein expression of calcineurin, MEF2, and NFATc1 in skeletal muscles. MEF2 and NFATc1 are important transcription factors for skeletal muscle fiber switching [41]. When translocated from the cytoplasm to the nucleus, NFAT regulated calcium-dependent target genes that promoted the formation of slow muscle fibers [42]. Another important muscle remodeling pathway is $Ca^{2+}$/CaMK, which increases MEF2, thereby promotes the formation of slow-twitch fiber types [43,44]. $Ca^{2+}$ played a dominant role in these two

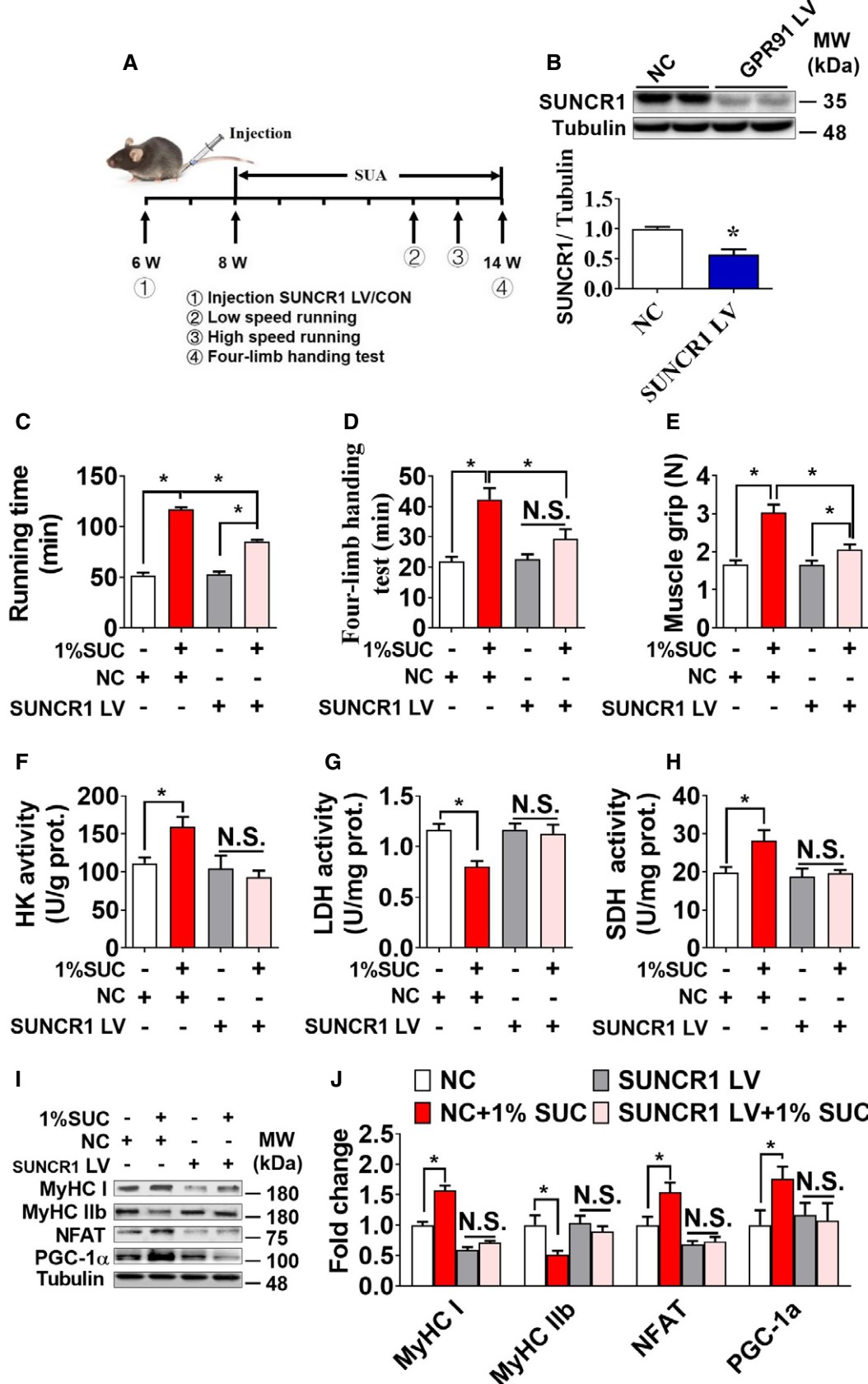

Figure 8.

◄

**Figure 8.   Gastrocnemius-specific SUNCR1 knockdown abolishes the effect of succinate on muscle fiber switch *in vivo*.**
Male C57BL/6J mice were injected with LV-shScrambled or shSUNCR1 lentivirus specifically into the gastrocnemius at 6 weeks of age. After 2 weeks of recovery, mice were fed with chow diet supplemented with 0 or 1% SUC for 6 weeks.

A       Timeline of the experimental protocol.
B       SUNCR1 protein expression in gastrocnemius from mice transfected with shSUNCR1 lentivirus or LV-shScrambled ($n = 3$).
C–E     (C) The running time in low speed, (D) four-limb handing time, and (E) muscle grip strength of both control and gastrocnemius-specific SUNCR1 knockdown mice.
F–H     The enzymes activity of (F) HK, (G) LDH, and (H) SDH in gastrocnemius.
I, J    Immunoblots and quantification of MyHC I, MyHC IIb, NFAT, and PGC-1α protein in gastrocnemius ($n = 3$).

Data information: Results are presented as mean $\pm$ SEM ($n = 5$–8). *$P \leq 0.05$ by non-paired Student's *t*-test.

signaling pathways [45]. Thus, we wondered if $Ca^{2+}$ mediated succinate-induced fiber-type switch in muscle.

To test this hypothesis, we blocked $[Ca^{2+}]_i$ by inhibiting PLC-β and found that succinate-induced fiber-type transition was effectively abolished by PLC-β antagonist. These results demonstrated that succinate-induced muscle fiber transition was closely associated with calcium signaling pathway and its downstream transcript factors, MEF2 and NFATc1. On the other hand, we found that succinate decreased p-AMPK/AMPK ratio, suggesting an increased intracellular energy state. The decreased AMPK activity might be attributed to the enhanced oxidative capacity and ATP production. This evidence indicated that AMPK signaling pathway might not be involved in succinate-induced skeletal muscle fiber-type transition.

Besides acting as a metabolite in the TCA cycle, succinate also exhibits a hormone-like function through the activation of G-protein-coupled receptor SUNCR1 [46]. SUNCR1 is expressed throughout the whole body [47,48] and has been reported to couple with either Gi or Gq protein to trigger different intracellular pathways [49]. For example, succinate elevates the levels of hemoglobin, platelets, and neutrophils [50] and enhances immunity [51] through SUNCR1-coupled $G_i$; it also increases intracellular calcium [52] coupled with Gq to release arachidonic acid along with prostaglandins E2 and I2. Here, we showed that succinate increased the expression of SUNCR1 and its downstream factor PLCβ, which were associated with boosted $[Ca^{2+}]_i$. This finding suggests that succinate may act on Gq-coupled SUNCR1 in skeletal muscles. In supporting this view, SUNCR1 global knockout or selective knockdown in skeletal muscle abolished the regulatory effects of succinate on muscle fiber transition both *in vitro* and *in vivo*. Our data demonstrated that SUNCR1 is the primary mediating receptor for the effect of succinate on skeletal muscle fiber-type remodeling.

Consistent with our previous report on the stimulatory effects of succinate on protein synthesis in skeletal muscle [23], we also found dietary supplementation of succinate activated Akt/mTOR cascade and inhibited FoxO3a in WT mice. These regulatory effects of succinate were diminished in SUNCR1 KO mice, suggesting a SUNCR1-mediated activation on protein synthesis. In this context, a seemingly paradoxical finding is that dietary supplementation of succinate failed to increase muscle mass. How can succinate increases skeletal muscle protein synthesis without changing muscle weight? We speculate that this inconsistency may be due to succinate-induced muscle type remodeling from fast- to slow-twitch fibers. It is known that slow-twitch fibers have lower fiber size and higher oxidative proteins and capacity for protein synthesis compared to fast-twitch fibers [53]. Succinate-induced hypertrophy of skeletal muscle may be neutralized by the discrepancy in fiber size of slow- and fast twitch or mass of large myofibrillar proteins and much smaller oxidative proteins. Alternatively, it is also

possible that the protein synthesis is balanced by a high rate of protein degradation resulting in a higher turnover rate in the high oxidative fibers.

Regular exercise and chronic hypoxia are natural stimuli that produce sustainable cardioprotection against ischemia reperfusion [54]. Consistent with the important role of succinate in muscle metabolism and fiber remodeling we showed, succinate is elevated in the blood in response to exercise [32] and accumulated rapidly in hypoxic/ischemic tissues [33,55,56], suggesting a potential role of succinate in exercise/hypoxia-mediated cardioprotection. Succinate may act as a paracrine or endocrine signaling molecules via SUCNR1 to regulate local cellular metabolism [57], or increase tissue blood supply through the renin-angiotensin system, thereby alleviating tissue hypoxia and hypoxia adaptation of metabolism in the environment [58–60]. Consistently, augmentation of succinate has been shown to improve cardiac ischemic energetics, a source of damage at reperfusion [55]. Therefore, succinic acid may not only play an important role in autocrine regulation of skeletal muscle metabolism and fiber-type conversion, but also improve the adaptability of cardiovascular and brain tissues to the ischemic environment.

Our results demonstrated that dietary succinate supplementation led to remodeling of muscle fiber without changing body weight or fat distribution, suggesting that the primary function of succinate is to regulate muscle type transition but not body weight. However, our study was carried out under normal chow diet (low-fat diet), which may have concealed a phenotype relevant for human obesity normally induced by high-energy/fat diet. Indeed, a recent study has shown that water supplementation of 1.5% but not 1% succinate stimulates uncoupling protein 1 (UCP1)-dependent thermogenesis from BAT, which induces robust protection against HFD-induced obesity [22]. This discrepancy suggests a diet-dependent anti-obesity effect of succinate, which may be attribute to different baseline UCP1 activation in chow and HFD condition. It has been shown that HFD significantly inhibits the expression and metabolic activity of UCP-1 in BAT [61]. The inconsistency may also be due to different supplementary method and dose (1.5% in water vs. 1% diet). The effective dose of succinate to remodel skeletal muscle fiber type may be lower than that to reduce body weight and fat mass.

In conclusion, our results demonstrated that succinate induces a SUNCR1-mediated transformation from fast- to slow-twitch fiber types in skeletal muscle. This finding indicates the potential application of succinate as exercise mimetics for people who are bedridden or disable to maintain their fitness, and even for athletes to improve their performance. Additional studies are warranted to identify the high-affinity ligands of SUNCR1, which may be helpful to maintain muscle energy homeostasis and alleviate fatigue.

# Materials and Methods

### Animal experiments

C57BL/6J about 3-week-old mice were purchased from the Medical Experimental Animal Center of Guangdong Province (Guangzhou, Guangdong, China). All animals raised and experiments were permitted by the College of Animal Science, South China Agricultural University, and in line with "the instructive notions with respect to caring for laboratory animals" issued by the Ministry of Science and Technology of the People's Republic of China. C57BL/6J mice were housed in an individual cage under the controlled room temperature ($23°C \pm 3°C$) and relative humidity ($70 \pm 10\%$) conditions, with a 12-h–12-h light–dark cycle. C57BL/6J mice were left to acclimate for 1 week, then randomly divided into three groups ($n = 11$) based on their body weight. Three groups of mice were fed with normal standard diets containing 0, 0.5%, or 1% succinic acid sodium salt, respectively. Body weight and food intake were measured every Monday in morning. Low-speed running was tested in the fourth week, fast running was tested in the fifth week, and four-limb handing test was tested in the sixth week. After 8 weeks, the mice were sacrificed and whole blood, serum, gastrocnemius, soleus, fat, and liver tissues were collected.

### UPLC–Orbitrap–MS/MS analysis

LC-MS/MS was performed as previously described [62]. In brief, chromatographic separation was performed on a C18 Hypersil Gold ($100 \times 2.1$ mm, 1.9 μm, Thermo Scientific) column using acetonitrile (eluent A) ultrapure and water-0.1% formic acid solution (eluent B) as mobile phase at a flow rate of 0.2 ml/min. The gradient program was set as follows: A 0–7 min, 5–50%; 7–8 min, 50–75%; 9–11 min, 80–90%; 11–15 min, 90–95%; and 15–20 min, 95%, with a total running time of 20 min. The column temperature was 35°C, and the injection volume was 2 μl. The MS data were acquired using electrospray ionization (ESI) in the negative and positive ionization modes, spray voltage, 4 kV (−4 kV in ESI−), 3.5 kV (+3.5 kV in ESI+); sheath gas ($N_2$, > 95%), 40 bar; auxiliary gas ($N_2$, > 95%), 10 bar; heater temperature, 300°C; and capillary temperature, 320°C. MS Scanning mode: Full MS scan ranged from m/z 100 to 1,500, and the resolution was 35,000; in-source collision-induced dissociation (in-source CID) was set at 0 eV. MS/MS scanning mode: Data-dependent $ms^2$ scan (dd-$ms^2$) with the resolution was 17,500, and high collision-induced dissociation (HCD) was set as stepped mode (10, 30, 50 eV). The test method is based on the paper of Xin et al and is slightly modified.

### Generation of Sucnr1 knockout mouse model

The SUCNR1 knockout mouse model used in this study was designed and developed by Shanghai Model Organisms Center, Inc (Shanghai, China). Briefly, Cas9 mRNA was in vitro-transcribed with mMESSAGE mMACHINE T7 Ultra Kit (Ambion, TX, USA) according to the manufacturer's instructions, linearized using NotI (NEB, USA), and subsequently purified using the MEGAclearTM Kit (Thermo Fisher, USA). Four independent sgRNAs designed to delete exon 2 of Sucnr1 were ①: 5′-TGGACCTTCAATACGAGGGC-3′, ②: 5′-GGCATGGACCTTCAATACGA-3′, ③: 5′-CGCCCATGATTAAATTCCAC-3′, and ④: 5′-GATCCTGTGGAATTTAATCA-3′. The sgRNAs

were transcribed in vitro using the MEGAshortscript Kit (Thermo Fisher, USA). In vitro-transcribed Cas9 mRNA and sgRNAs were injected into zygotes of C57BL/6J mouse and transferred to pseudo-pregnant recipients. Obtained F0 mice were screened by PCR and sequencing using primer pairs: F1 5′-GGCTGGCCATGAAGATACA-3′; R1 5′-TTGGAATGCCGAGAACTGAG-3′. The positive F0 mice were chosen and crossed with C57BL/6J mice to obtain F1 heterozygous Sucnr1 knockout mice. The genotype of F1 mice was identified by PCR and confirmed by sequencing. Male and female F1 heterozygous mice were intercrossed to produce the homozygous Sucnr1 knockout mice.

### In vivo oxygen consumption assay

After mice were fed with different concentration of succinate for 6 weeks, $O_2$ consumption ($VO_2$) and respiratory exchange ratio (RER) were obtained using the promotion metabolism measurement system (Sable Systems International, USA).

### Strength and exercise endurance

Mouse had maximum muscle force measured three times by a grip strength meter (BIO-GS3, Bioseb/France), and the mean maximum strength was used for data analysis. The mice performed a treadmill-running test on the FT-200 Animal treadmill at an initial velocity of 10 m/min for 10 min in order to keep mice sober. Then, velocity was increased by 5 m/min every 2 min until 40 m/min in high-speed running tests, and 1 m/min every 3 min in low-speed running tests. The above tests refer to the previous study [63]. Inverted screen, which was a 43 $cm^2$ of wire mesh, consisted of 12 $mm^2$ of 1-mm-diameter wire, made as the previous article [64] to test mice falling time. Fat mass, lean mass, and body composition were determined using a nuclear magnetic resonance system (Body Composition Analyzer MiniQMR23-060H-I, Niumag Corporation, Shanghai, China).

### Ex vivo gastrocnemius muscle force and fatigability measurements

For ex vivo gastrocnemius fatigability assessments, mice were anesthetized, and the gastrocnemius muscles along with their tendons were surgically removed and then kept in an aerated bath of physiological Krebs solution. Each muscle was mounted vertically in a double-jacketed bath of aerated (95% $O_2$/5% $CO_2$) physiological solution (2.5 mM $Ca^{2+}$ tyrode solution: 140 mM NaCl, 5 mM KCl, 10 mM HEPES, 2.5 mM $CaCl_2$, 2 mM $MgCl_2$, and 10 mM glucose) at room temperature. Supramaximal electricity with a pulse width of 1 ms was delivered to muscles by a pair of platinum electrodes placed in parallel. Following equilibration of the muscle, several baseline twitches were recorded. Muscles were subjected to an intermittent stimulation protocol in which a stimulus train at 180 times per minute was administered. Intermittent fatigue stimulation was used to test fatigue resistance for 80 s. The contractile performance was assessed by measuring half relaxation time (the time required for force to decrease 50% from the peak value at the end of stimulation). The ex vivo contractility experiment was set up using the BL-420F biological signal acquisition and analysis system (Chengdu Taimeng software Co., Ltd. China) [65].

## Cell culture

The mouse myoblast cell line C2C12 (ATCC) was cultured in high glucose DMEM (GIBCO, Grand Island, NY, USA) containing 10% fetal bovine serum (FBS), 100 U/ml penicillin, and 100 μg/ml streptomycin at 37°C, in a humidified atmosphere containing 5% $CO_2$. When cells reached 90% confluency, culture media was switched by DMEM with 2% horse serum to induce myoblasts' differentiation to myotubes for 6 days.

## Immunofluorescence staining and imaging

For staining of muscle sections, we collected mouse gastrocnemius muscle samples frozen by liquid nitrogen-cooled isopentane in Tissue-Tek OCT and then sliced muscles into 10 μm by a cryostat (CM1850, Leica) for staining. Muscle sections were fixed with paraformaldehyde (PFA)/PBS (1%, 10 min), quenched with glycine (50 mM, 10 min), permeabilized with Triton X-100 (0.5%, 10 min), blocked with Mouse On Mouse (M.O.M.) Blocking Reagent (Vector Laboratories) and 5% BSA/5% normal goat serum/PBS, and incubated with primary antibodies. Antibodies used included mouse anti-MyHC I (BA-D5-S 1:100, DSHB), mouse anti-MyHC IIb (BF-F3 1:100, DSHB), and rabbit anti-laminin (PA1-16730 1:1,000, Thermo Fisher). Sections were washed in PBS/0.1% Tween-20, incubated with Alexa Fluor-labeled (goat Anti-Mouse IgM/Alexa Fluor 555 antibody, bs-0368G-AF555, Bioss) and FITC-labeled (goat anti-mouse FITC, bs-50950, Bioworld) secondary antibodies (1:1,000, 1 h). Mounted slides were imaged on a LEICA TCS SP8 (LEICA, Germany) confocal microscope.

For staining of C2C12 cell, C2C12 cell was punched in 0.4% Triton for 10 min and then blocked for 1 h in slowly shaking at room temperature. The sections were then immunostained with primary antibody at room temperature overnight in a wet box. Goat anti-rabbit FITC (bs-0295G, Bioss), goat anti-mouse IgM/Alexa Fluor 555 antibody (bs-0368G-AF555, Bioss), goat anti-rabbit Flour 555 (bs-0295G, Bioss), goat anti-mouse FITC (bs-50950, Bioworld), rabbit anti-goat IgG FITC (bs-0294R, Bioss), and corresponding second antibodies were supplied for use. A Nikon Eclipse Ti-s microscope was used to take photos of these sections. Images of fluorescent intensity were captured with Nis-Elements BR software (Nikon Instruments, Tokyo, Japan).

## Quantification of muscle cross-sectional area

In cryosectioned muscle preparations, we used immunofluorescence for laminin staining. All measurements were made by a single person blinded to the hypothesized outcomes. To determine the relative size of muscle fibers, we measured muscle fiber cross-sectional area (CSA) and fiber perimeter in skeletal muscle. Each group measured the same number of skeletal muscle sections. Muscle fiber cross-sectional area was determined using MetaMorph software (image pro plus 6.0, MEDIA CYBERNETICS, United States).

## Succinate dehydrogenase staining

Succinate dehydrogenase staining was performed as previously described [66]. Briefly, muscle sections (10 μM) were incubated in liquid (6 mM $CaCl_2$, 0.3% glacial acetic acid, pH 4.4) for 10 min,

flushed by Tris-$CaCl_2$ eluent buffer (0.1 M Tris, 18 mM $CaCl_2$) twice, 1 min per flush, and then incubated in 37°C pre-heated SDH eluent buffer (0.1 M sodium succinate, 0.18 mM tetranitroblue tetrazolium chloride (NBT), 0.81% N,N-dimethylformamide, 0.23 M Tris, pH 7.4) for 45 min. The sections were washed with distilled water twice and then incubated in 37°C pre-heated ATPase eluent buffer (3 mM adenosine 5′-triphosphate disodium salt, 0.2 M Tris, 18 mM $CaCl_2$, 50 mM KCl, pH 9.4) for 30 min. Sections were then washed with distilled water twice, incubated in 2% $CoCl_2$ for 4 min, and washed carefully with distilled water twice. Sections were then incubated in 2% ammonium sulfide for 30 s, followed by careful washing with distilled water, twice. After staining with Ehrlich's hematoxylin, sections were sent for dehydration in alcohol and fixated by neutral balsam. Images were captured with an olympus CX41 microscope (Olympus Corporation, Japan). Four different horizontal regions were captured in each section, and images were acquired with Meta-Morph software (image pro plus 6.0, MEDIA CYBERNETICS, United States) for morphology measurements. The amount of SDH staining in the four horizontal regions was analyzed.

## [Ca²⁺]i assay

$[Ca^{2+}]i$ was measured by calcium fluorometry following the manufacturer's instructions of fluo-8 AM kit. After induced into myotubes, C2C12 cells were washed twice with Hank's Balanced Salt Solution (HBSS, pH = 7.2–7.4) containing 8 g/l NaCl, 0.4 g/l KCl, 0.1 g/l $MgSO_4 \cdot 7H_2O$, 0.1 g/l $MgCl_2 \cdot 6H_2O$, 0.06 g/l $Na_2HPO_4 \cdot 2H_2O$, 0.06 g/l $KH_2PO_4$, 1 g/l glucose, 0.14 g/l $CaCl_2$, and 0.35 g/l $NaHCO_3$, and incubated with 10 μM fluo-8-AM at 37°C for 1 h. After incubation, cells were then washed twice again. Nikon Eclipse Ti-s microscopy was used to observe fluorescence which was initiated by succinate. Fluorometric data were acquired at excitation and emission wavelengths of 490 and intensity at 525 nm (490/525 nm), for every 2-s interval over a 180-s period.

## Measurement of oxygen consumption rate (OCR) in gastrocnemius tissue homogenate

All measurements were done using a high-resolution respirometer (Oxygraph-2k, Oroboros Instruments, Innsbruck, Austria). Before the experiments, the Oxygraph was calibrated to correct for back diffusion of oxygen into the chamber, leak from the exterior, oxygen consumption by the chemical medium, and by the polarographic oxygen sensor. $O_2$ flux was resolved by software (Datlab 5, Oroboros Instruments, Innsbruck, Austria). All respirometry measurements were done in duplicate in the respiration medium MiR05 (110 mmol/l sucrose, 60 mmol/l potassium lactobionate, 0.5 mmol/l EGTA, 3 mmol/l $MgCl_2 \cdot 6H_2O$, 20 mmol/l taurine, 10 mmol/l $KH_2PO_4$, 20 mmol/l HEPES, 1 g/l BSA, pH 7.1 at 37°C) at 37°C after hyperoxygenation (450–200 nmol/ml) to avoid oxygen limitations.

## SUNCR1 siRNA transfection

The transfection steps and siRNA sequences of SUNCR1 were described in our previous study [67]. The siRNA of SUNCR1 was purchased from GenePharma Co., Ltd (Shanghai, China) and transfected with lipofectamine (Invitrogen, Carlsbad, CA, USA) in accordance with the manufacturer's instructions.

## SUNCR1 knockdown

The shSUNCR1 lentivirus was generated from Hanbio Biotechnology Co., Ltd (Shanghai, China). Thirty-two mice were randomly divided into four groups ($n = 8$): LV-shScrambled, LV-shScrambled+SUA, LV-SUNCR1, and LV-shSUNCR1 + SUA. After the interference efficiency was verified, 60 μl ($10^7$ titers) lentivirus was intramuscularly injected in three different sites of the gastrocnemius.

## Mitochondrial staining

The mitochondrial staining was performed by using Mito-Tracker Green (C1048) purchased from Beyotime Biotechnology Institute (China). Initially, Mito-Tracker Green was formulated with anhydrous DMSO (anhydrous dimethyl sulfoxide) to the concentration of 1 mM, while the working concentration is 100 nM diluted with DMEM, incubating cells for 30 min at 37°C. The cells were then washed twice with phosphate-buffered saline (PBS). Pictures were taken and analyzed by Nikon Eclipse Ti-microscope and Nis-Elements BR software. The work of mitochondrial electron microscopy was done by Fucheng Biotechnology Institute (China).

## Mitochondrial electron microscopy

C2C12 cells were gathered from a cell culture dish and preserved in 5% glutaraldehyde, and diluted with phosphate buffer for at least 2 h. The cells were dissected into 1 mm³ and carefully washed in phosphate rinse solution for 15 min (three times). Cells were post-fixed in 1% osmium tetroxide solution for 2–3 h and carefully washed in phosphate rinse solution for 15 min (three times), then dehydrated with increasing concentrations of ethanol. Cells were incubated in acetone and solidified in the oven. Ultrathin sectioning was then sliced by Ultra Microtome Leica UC6 in 70 nm and collected in grids. 3% uranyl acetate-lead citrate double-stained the grids. Images were obtained from a Jeol1230 transmission electron microscope at 120 kV at ×10,000, ×20,000, and ×50,000 magnification for posterior analysis.

## MitoProbe™ TMRM assay

For each sample, the cells were re-suspend in cell culture medium or PBS at approximately $1 \times 10^6$ cells/ml. For the control samples, 1 μl of 50 mM CCCP was added to the cells and incubated for 5 min at 37°C, 5% $CO_2$. Experimental samples had 1 μl of 20 μM stock TMRM (M20036, Thermo Scientific) reagent (20 nM final concentration) added and were incubated for 30 min at 37°C. Cells were washed once in 1 ml of PBS and then re-suspended in 500 μl of PBS. The cells were analyzed on a CytoFLEX software (Beckman Coulter, USA) with 561-nm excitation, using emission filters appropriate for R-phycoerythrin.

## Mitochondrial DNA

Total cellular DNA was extracted from C2C12 cells with DNAzol reagent (Invitrogen, CA, USA) according to the manufacturer's instructions. Mitochondrial DNA copy number was determined by quantification of four mitochondrial marker genes, including mitochondrially encoded ATP synthase membrane subunit 6 (ATPase6), cytochrome c oxidase subunit 2 (COX2), Mit-1000, and mitochondrial-encoded cytochrome b (mt-Cytb). The expression level of ATPase6, COX2, Mit-1000, and mt-Cytb was tested by quantitative real-time–PCR and normalized to an intron of the nuclear-encoded β-globin gene as described before [68,69]. The primer sequences can be found in the Table EV1.

## Western blot assay

We use RIPA lysis buffer containing 1 mM PMSF to lyse C2C12 cell or muscles. For the nuclear or cytoplasmic protein extraction, proteins were isolated according to the procedure of the nuclear extraction kit (Solarbio, SN0020). Protein concentration was determined using a BCA protein assays kit. After sodium dodecyl sulfate (SDS)–polyacrylamide gel electrophoresis gels, primary antibodies were used, including rabbit anti-β-tubulin (bs-1482M, 1:5,000, Bioss), rabbit anti-SUNCR1 (NBP1-00861, 1:1,000, Novus), mouse anti-MyHC I (ab11083, 1:1,000; Abcam), rabbit anti-MyHC IIa (ab124937, 1:1,000, Abcam), goat anti-MyHC IIb (sc-168672, 1:500; Santa Cruz), mouse anti-PGC-1α (ST1202, 1:1,000, Millipore), rabbit anti-histone (4499S, 1:2,000; CST), mouse anti-NFAT (sc-7294, 1:500; Santa Cruz), rabbit anti-NRF-1 (#12381s, 1:2,000, CST), rabbit anti-calcineurin (#2614s, 1:2,000; CST), rabbit anti-Myoglobin (ab77232, 1:1,000, Abcam), and rabbit anti-MEF2A (#97365, 1:2,000; CST). Protein expression levels were determined using MetaMorph software (ImageJ, National Institutes of Health, USA).

## RNA extraction, reverse transcript, and qPCR

We extracted total RNA from C2C12 cell lines using an RNA extraction kit (Guangzhou Magen Biotechnology Co., Ltd, China). Skeletal muscles were dissolved in TRIzol reagent (Invitrogen, Carlsbad, CA, USA) according to the manufacturer's instructions. 2 μg of total RNA was treated with DNase I (Takara Bio Inc., Shiga, Japan) and reverse transcribed to cDNA by M-MLV Reverse Transcriptase (Promega, Madison, WI, USA) and random primers 9 (Takara Bio Inc., Shiga, Japan) according to the manufacturer's instructions. cDNA synthesis was performed with the Applied Biosystems QuantStudio 3 Real-Time PCR System (Thermo Fisher Scientific, USA).

## Metabolites and enzyme activities assay

Triglyceride (TG), non-esterified fatty acid (NEFA), lactic acid (LD), the activity of lactic dehydrogenase (LDH), succinodehydrogenase (SDH), and hexokinase (HK) were all measured by commercial assay kits which were purchased from Nanjing Jiancheng Bioengineering Institute (China).

## Statistical analysis

All data are presented as means ± the standard error of the mean (SEM). The difference between control and dose-effect groups was determined by one-way ANOVA tests (GraphPad Prism 6.0). $P < 0.05$ was considered statistically significant.

**Expanded View** for this article is available online.

## Acknowledgements

This work was supported by National Key Point Research and Invention Program (2018YFD0500403 to G. S.), National Natural Science Foundation of China (31790411 to Q. J. and 31572480 to G. S.), National Key Point Research

and Invention Program (2016YFD0501205 to G. S.), Guangdong Key areas Research and Development Project (2019B020218001 to G. S), National Institutes of Health (K99DK107008 to P.X.), and Innovation Team Project in Universities of Guangdong Province (2017KCXTD002 to G. S.)

## Author contributions

TW, Y-QX, Y-XY, X-CC, CZ, J-RX, P-WX, DY, Z-RL, L-LY, LL, GZ, and B-CD carried out all experimental work; CJZ, L-NW, FL, CY, LZ, J-PY, MD, Y-PZ, S-BW, X-TZ, Q-YX, PG, Y-LZ, and Q-YJ conducted part of cell culture, animal studies, Western blot, qPCR and other and data analysis; GS designed this experiment; Z-HH, B-QL, Y-QX, P-PX, TW, SS, and GS contributed to article preparation.

## Conflict of interest

The authors declare that they have no conflict of interest.

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
