## [Review Process File · EMBO Reports]

Succinate induces skeletal muscle fiber remodeling via SUNCR1 Signaling Pathway

Tao Wang, Ya-Qiong Xu, Ye-Xian Yuan, Ping-Wen Xu, Cha Zhang, Fan Li, Li-Na Wang, Cong Yin, Lin Zhang, Xing-Cai Cai, Can-Jun Zhu, Jing-Ren Xu, Bing-Qing Liang, Sarah Schaul, Pei-Pei Xie, Dong Yue, Zheng-Rui Liao, Lu-Lu Yu, Lv Luo, Gan Zhou, Jin-Ping Yang, Zhi-hui He, Man Du, Yu-Ping Zhou, Bai-Chuan Deng, Song-Bo Wang, Ping Gao, Xiao-Tong Zhu, Qian-Yun Xi, Yong-Liang Zhang, Gang Shu and Qing-Yan Jiang

Review timeline:

Submission date:	7 February 2019
Editorial Decision:	15 March 2019
Revision received:	10 May 2019
Editorial Decision:	7 June 2019
Revision received:	13 June 2019
Accepted:	26 June 2019

Editor: Deniz Senyilmaz-Tiebe

Transaction Report:

1st Editorial Decision

15 March 2019

Thank you for submitting your manuscript for consideration by EMBO Reports. It has now been seen by two referees whose comments are shown below.

As you can see, both referees express interest in your study demonstrating that succinate activates muscle remodeling. However, they also raise concerns that need to be addressed in full before we can consider publication of the manuscript here.

Given these constructive comments, I would like to invite you to revise your manuscript with the understanding that the referee must be fully addressed and their suggestions taken on board. Please address all referee concerns in a complete point-by-point response. Of note, please address the point 4 of referee #1 by measuring mitochondrial membrane potential with another method such as TMRE staining rather than omitting the data. Acceptance of the manuscript will depend on a positive outcome of a second round of review. It is EMBO Reports policy to allow a single round of revision only and acceptance or rejection of the manuscript will therefore depend on the completeness of your responses included in the next, final version of the manuscript.

Supplementary/additional data: The Expanded View format, which will be displayed in the main

HTML of the paper in a collapsible format, has replaced the Supplementary information. You can submit up to 5 images as Expanded View. Please follow the nomenclature Figure EV1, Figure EV2 etc. The figure legend for these should be included in the main manuscript document file in a section called Expanded View Figure Legends after the main Figure Legends section. Additional Supplementary material should be supplied as a single pdf labeled Appendix. The Appendix includes a table of content on the first page with page numbers, all figures and their legends. Please follow the nomenclature Appendix Figure Sx throughout the text and also label the figures according to this nomenclature. For more details please refer to our guide to authors.

When preparing your letter of response to the referees' comments, please bear in mind that this will form part of the Review Process File, and will therefore be available online to the community. For more details on our Transparent Editorial Process, please visit our website:
http://emboj.embopress.org/about#Transparent_Process

Regarding data quantification, please ensure to specify the name of the statistical test used to generate error bars and P values, the number (n) of independent experiments underlying each data point (not replicate measures of one sample), and the test used to calculate p-values in each figure legend. Discussion of statistical methodology can be reported in the materials and methods section, but figure legends should contain a basic description of n, P and the test applied. Please also include scale bars in all microscopy images.

We now strongly encourage the publication of original source data with the aim of making primary data more accessible and transparent to the reader. The source data will be published in a separate source data file online along with the accepted manuscript and will be linked to the relevant figure. If you would like to use this opportunity, please submit the source data (for example scans of entire gels or blots, data points of graphs in an excel sheet, additional images, etc.) of your key experiments together with the revised manuscript. Please include size markers for scans of entire gels, label the scans with figure and panel number, and send one PDF file per figure.

- a complete author checklist, which you can download from our author guidelines (<http://embor.embopress.org/authorguide#revision>). Please insert page numbers in the checklist to indicate where the requested information can be found.
 - a letter detailing your responses to the referee comments in Word format (.doc)
 - a Microsoft Word file (.doc) of the revised manuscript text
 - editable TIFF or EPS-formatted figure files in high resolution
- (In order to avoid delays later in the publication process please check our figure guidelines before preparing the figures for your manuscript:
http://www.embopress.org/sites/default/files/EMBOPress_Figure_Guidelines_061115.pdf)
- a separate PDF file of any Supplementary information (in its final format)
 - all corresponding authors are required to provide an ORCID ID for their name. Please find instructions on how to link your ORCID ID to your account in our manuscript tracking system in our Author guidelines (<http://embor.embopress.org/authorguide>).

As part of the EMBO publication's Transparent Editorial Process, EMBO reports publishes online a Review Process File to accompany accepted manuscripts. This File will be published in conjunction with your paper and will include the referee reports, your point-by-point response and all pertinent correspondence relating to the manuscript.

I look forward to seeing a revised version of your manuscript when it is ready. Please let me know if you have questions or comments regarding the revision.

REFeree REPORTS**Referee #1:**

Overview

This is an interesting paper that provides strong evidence for the role of circulating succinate in activating muscle fiber remodelling through the SUCNR1 receptor. In general the work is of high quality and the data support the interpretation. Overall, I think this is a solid paper that adds considerably to an emerging field. However, I do have some comments and technical points that I feel will require more experiments.

Major points

1 I think that since GPR91 was de-orphaned that it is now called SUCNR1.

2 I don't believe the data in Fig 1A - assuming SUA means succinic acid? The control concentration of succinic acid is about 85 mg/l. As the Mw of succinic acid is 118, this is a concentration of 720 μ M!! The normal serum concentration of succinate is about 1 - 2 μ M (eg see Journal of the American Heart Association (2018) 7 e007546). The use of the Sigma kit for plasma and tissue is not acceptable and this needs to be redone properly by LC-MS.

3 The food is labelled as 0.5% or 1% succinate but assuming this is a salt then the counterion isn't defined.

4 The use of JC1 to assess changes in mitochondrial membrane potential is not a good method as it is artifact prone and not quantitative. I suggest repeating with a more reliable method (eg TMRM) or omitting the data.

5 The analysis of mtDNA does not seem to have been normalised to a nuclear gene? Without that the data are hard to interpret and should be redone.

6 A major source of succinate seems to be ischemic tissue which accumulates succinate (see Nature 2014 515 431-435) and releases it into the circulation Journal of the American Heart Association (2018) 7 e007546; Cell Reports 2018 23 2617). This should be discussed and its possible link to SUCNR1 activation.

7 In the recent Nature paper in 2018 it was shown that succinate activated BAT and led to weight loss. The reasons why you don't see weight loss here should be discussed.

Minor points

1 SUA (succinate acid?) wasn't defined in the abbreviations.

2 The paper was well written and logically constructed, but in a few places there were typos and nonidiomatic phrases - a quick edit by a native speaker would be helpful.

Referee #2:

In the manuscript Titled "Succinate Induces skeletal muscle fiber remodeling via GPR91 Signaling Pathway" the authors analyzed the effect of succinate increases in endurance exercise ability, slow vs. fast-twitch fiber types markers, aerobic enzyme activity, oxygen consumption and mitochondrial biogenesis in skeletal muscle.

In addition, the authors show evidence for a GPR91 role in mediating succinate effect on skeletal muscle fiber type remodeling suggesting a potential use of succinate-based compounds in both athletic and sedentary populations.

Overall, the work is might provide valuable information regarding the role of succinate in fast/slow

muscle fiber twitch and their metabolism.

However, in order to be suitable for publication, the manuscript deserves further experiments.

Major points:

1. In 2017 the same group published the following paper: Succinate promotes skeletal muscle protein synthesis via Erk1/2 signaling pathway from the same group Yexian Yuan, et al. published in Mol Med Rep. 2017 Nov; 16(5): 7361-7366. In this paper the authors provide evidence that succinate stimulates protein synthesis along with the increase of the Akt/mTOR/FoxO pathway. How the authors explain the discrepancy between the observed succinate-mediated protein synthesis (Yexian Yuan, et al.) and the data presented in the present paper regarding the lack of increased muscles weight?

In the present work, the authors should include a biochemical analysis of the Akt/mTOR/FoxO pathway on WT not treated and in vivo fed with succinate-supplemented chow and in GPR91 KO mice.

2. Since the Gastrocnemius is a mixed muscle, the authors should include evidences in EDL (fast/fast) and soleus (slow/slow) muscles.

3. To assess the distribution of the slow vs. fast fibers, the authors should perform a metachromatic staining (NADH or SDH staining).

1st Revision - authors' response

10 May 2019

Referee #1:

Overview

This is an interesting paper that provides strong evidence for the role of circulating succinate in activating muscle fiber remodelling through the SUCNR1 receptor. In general the work is of high quality and the data support the interpretation. Overall, I think this is a solid paper that adds considerably to an emerging field. However, I do have some comments and technical points that I feel will require more experiments.

Major points

1. I think that since GPR91 was de-orphaned that it is now called SUCNR1.

This point is well taken. GPR91 has been replaced by SUCNR1 in this manuscript revision.

2 I don't believe the data in Fig 1A - assuming SUA means succinic acid? The control concentration of succinic acid is about 85 mg/l. As the Mw of succinic acid is 118, this is a concentration of 720 μM!! The normal serum concentration of succinate is about 1-2 μM (eg see Journal of the American Heart Association (2018) 7 e007546). The use of the Sigma kit for plasma and tissue is not acceptable and this needs to be redone properly by LC-MS.

We highly appreciate this point. The mouse serum succinic acid concentration was re-measured by LC-MS and Fig 1A was replaced by new results. The control serum succinic acid level is around 0.4 ng/μL (mg/L), which is equal to 3.4 μM and comparable to normal serum succinate in human patients (1-2 μM, Journal of the American Heart Association (2018) 7 e007546).

3 The food is labelled as 0.5% or 1% succinate but assuming this is a salt then the counterion isn't defined.

We appreciate the point. Succinic acid sodium salt was used for dietary supplementation. The following description has been added in line 589-590:

“Three groups of mice were fed with normal standard diets containing 0, 0.5% or 1% Succinic acid sodium salt, respectively”.

4 The use of JC1 to assess changes in mitochondrial membrane potential is not a good method as it is artifact prone and not quantitative. I suggest repeating with a more reliable method (eg TMRM) or omitting the data.

This is an excellent point. The mitochondrial membrane potential was re-tested by TMRM and supplementary Fig EV2B-C was replaced by new results.

5 The analysis of mtDNA does not seem to have been normalized to a nuclear gene? Without that the data are hard to interpret and should be redone.

Sorry for any confusion it may have caused. For mtDNA assay, we used β -globin as a nuclear reference gene for calibration. The detailed method description has been updated in the present version as following:

“Total cellular DNA was extracted from C2C12 cells with DNAzol reagent (Invitrogen, CA, USA) according to the manufacturer’s instructions. Mitochondrial DNA copy number was determined by quantification of four mitochondrial marker genes, including mitochondrially encoded ATP synthase membrane subunit 6 (ATPase6), cytochrome c oxidase subunit 2 (COX2), Mit-1000 and mitochondrial encoded cytochrome b (mt-Cytb). The expression level of ATPase6, COX2, Mit-1000 and mt-Cytb was tested by quantitative real-time PCR and normalized to an intro of the nuclear-encoded β -globin gene as described before [67,68]. The primer sequences can be found in the Supplementary Table 1”.

6 A major source of succinate seems to be ischemic tissue which accumulates succinate (see Nature 2014 515 431-435) and releases it into the circulation Journal of the American Heart Association (2018) 7 e007546: Cell Reports 2018 23 2617). This should be discussed and its possible link to SUNCRI activation.

This is an excellent point. The following discussion has been added.

“Regular exercise and chronic hypoxia are natural stimuli that produce sustainable cardioprotection against ischemia-reperfusion [53]. Consistent with the important role of succinate in muscle metabolism and fiber re-modeling we showed, succinate is elevated in the blood in response to exercise [32] and accumulated rapidly in hypoxic/ischemic tissues [33, 54, 55], suggesting a potential role of succinate in exercise/hypoxia-mediated cardioprotection. Succinate may acts as a paracrine or endocrine signaling molecules via SUCNR1 to regulate local cellular metabolism [56], or increase tissue blood supply through the renin-angiotensin system, thereby alleviating tissue hypoxia and hypoxia adaptation of metabolism in the environment [57-59]. Consistently, augmentation of succinate has been shown to improve cardiac ischemic energetics, a source of damage at reperfusion [60]. Therefore, succinic acid may not only play an important role in autocrine regulation of skeletal muscle metabolism and fiber type conversion, but also improve the adaptability of cardiovascular and brain tissues to the ischemic environment.”

7 In the recent Nature paper in 2018 it was shown that succinate activated BAT and led to weight loss. The reasons why you don't see weight loss here should be discussed.

This is an excellent point. We appreciate the suggestion and agree the necessity to discuss the discrepancy between the recent Nature paper (PMID: 30022159) and our results. The following discussion has been added.

“Our results demonstrated that dietary succinate supplementation led to remodeling of muscle fiber without changing body weight or fat distribution, suggesting that the primary function of succinate is to regulate muscle type transition but not body weight. However, our study was carried out under normal chow diet (low fat diet), which may have concealed a phenotype relevant for human obesity normally induced by high-energy/fat diet. Indeed, a recent study has shown that water supplementation of 1.5% but not 1% succinate stimulates uncoupling protein 1 (UCP1)-dependent thermogenesis from BAT, which induces robust protection against HFD-induced obesity [22]. This

discrepancy suggests a diet-dependent anti-obesity effect of succinate, which may be attributed to different baseline UCP1 activation in chow and HFD condition. It has been shown that HFD significantly inhibits the expression and metabolic activity of UCP-1 in BAT [61]. The inconsistency may also be due to different supplementary method and dose (1.5% in water vs 1% diet). The effective dose of succinate to remodel skeletal muscle fiber type may be lower than that to reduce body weight and fat mass.”

Minor points

1 SUA (succinate acid?) wasn't defined in the abbreviations.

SUA has been defined in FOOTNOTES.

2 The paper was well written and logically constructed, but in a few places there were typos and nonidiomatic phrases - a quick edit by a native speaker would be helpful.

Proofreading has been done by a native speaker.

Referee #2:

In the manuscript Titled "Succinate Induces skeletal muscle fiber remodeling via GPR91 Signaling Pathway" the authors analyzed the effect of succinate increases in endurance exercise ability, slow vs. fast -twitch fiber types markers, aerobic enzyme activity, oxygen consumption and mitochondrial biogenesis in skeletal muscle. In addition, the authors show evidence for a GPR91 role in mediating succinate effect on skeletal muscle fiber type remodeling suggesting a potential use of succinate-based compounds in both athletic and sedentary populations. Overall, the work is might provide valuable information regarding the role of succinate in fast/slow muscle fiber twitch and their metabolism. However, in order to be suitable for publication, the manuscript deserves further experiments.

Major points:

1. In 2017 the same group published the following paper: Succinate promotes skeletal muscle protein synthesis via Erk1/2 signaling pathway from the same group Yexian Yuan, et al. published in *Mol Med Rep.* 2017 Nov; 16(5): 7361-7366. In this paper the authors provide evidence that succinate stimulates protein synthesis along with the increase of the Akt/mTOR/FoxO pathway. How the authors explain the discrepancy between the observed succinate-mediated protein synthesis (Yexian Yuan, et al.) and the data presented in the present paper regarding the lack of increased muscles weight? In the present work, the authors should include a biochemical analysis of the Akt/mTOR/FoxO pathway on WT not treated and in vivo fed with succinate-supplemented chow and in GPR91 KO mice.

This is an excellent point. A biochemical analysis of AKT/mTOR/FOXo3a signaling in the gastrocnemius has been added in both WT and GPR91 KO in the current revision. We found a similar stimulatory effect of succinate on AKT/mTOR/FOXo3a pathway in WT (Figs. EV1C-D), which is blocked in GPR91 KO mice (Figs. EV4H-I), suggesting a GPR91-mediated activation on protein synthesis. We speculate that the discrepancy between increased protein synthesis and unchanged muscle mass may be due to succinate-induced muscle type remodeling. The following discussion has been added in the manuscript revision.

“Consistent with our previous report on the stimulatory effects of succinate on protein synthesis in skeletal muscle [53], we found dietary supplementation of succinate activated Akt/mTOR cascade and inhibited FoxO3a in WT mice. These regulatory effects of succinate were diminished in SUNCR1 KO mice, suggesting a SUNCR1-mediated activation on protein synthesis. In this context, a seemingly paradoxical finding is that dietary supplementation of succinate failed to increase muscle mass. How can succinate increases skeletal muscle protein synthesis without changing muscle weight? We speculate that this inconsistency may be due to succinate-induced muscle type remodeling from fast-twitch fibers to slow-twitch fibers. It is known that slow-twitch fibers have lower fiber size and higher oxidative proteins and capacity for protein synthesis compared to fast-

twitch fibers [54]. Succinate-induced hypertrophy of skeletal muscle may be neutralized by the discrepancy in fiber size of slow- and fast-twitch or mass of large myofibrillar proteins and much smaller oxidative proteins. Alternatively, it is also possible that the protein synthesis is balanced by a high rate of protein degradation resulting in a higher turnover rate in the high oxidative fibers.”

2. Since the Gastrocnemius is a mixed muscle, the authors should include evidences in EDL (fast/fast) and soleus (slow/slow) muscles.

This is an excellent point. We examined MyHC I and IIb protein expression in both soleus (SOL) and extensor digitorum longus (EDL) muscle by immunofluorescence after dietary supplementation of 0.5% or 1% succinate in WT mice (Figs. EV1K-N). We found that succinate dose-dependently increased MyHC I but not MyHC IIb protein expression in SOL, suggesting an increased proportion of slow-twitch fiber. On the other hand, succinate failed to affect the muscle fiber composition of EDL muscles.

3. To assess the distribution of the slow vs. fast fibers, the authors should perform a metachromatic staining (NADH or SDH staining).

We appreciate this constructive suggestion. Oxidative capacity of SOL, EDL and gastrocnemius muscles has been analyzed by succinate dehydrogenase (SDH) staining after dietary supplementation of 0.5% or 1% succinate in WT mice (Figs. EV1E-J). We found that succinate dose-dependently increased the percentage of SDH-positive fibers in SOL, EDL and gastrocnemius muscles, suggesting succinate is sufficient to improve mitochondrial content and oxidative capacity of mixed (gastrocnemius), slow/slow (SOL) or fast/fast (EDL) muscles.

2nd Editorial Decision

7 June 2019

Thank you for submitting the revised version of your manuscript. It has now been seen by both of the original referees.

As you can see, both referees find that the study is significantly improved during revision and recommend publication. Before I can accept the manuscript, I need you to address the below minor/editorial points:

Thank you again for giving us to consider your manuscript for EMBO Reports, I look forward to your minor revision.

REFeree REPORTS

Referee #1:

The authors have addressed all the points I raised to my satisfaction.

Referee #2:

In the revised manuscript titled "Succinate induces skeletal muscle fiber remodeling via SUNCRI Signaling Pathway" the authors address all the suggested comments made by the reviewers, making the work suitable for publication.

Minor points:

- This reviewer suggests re-ordering the figures in progressive order based on the mention in the text (i.e. Fig2D become Fig2A and Fig 2A become Fig 2D; FigEV1K-L and EV1M-N should be numbered before Figs. EV1E-J; Fig 5C is mentioned after fig 5F; Fig EV3B is mentioned after Fig. EV3C-E and so on.....). This will facilitate the reading.
- In Fig. EV1H the graph relative to the % of SDH positive fibers in Soleus do not correspond.

2nd Revision - authors' response

13 June 2019

Referee #2:

In Fig. EV1H the graph relative to the % of SDH positive fibers in Soleus do not correspond.
The graph relative to the % of SDH positive fibers in Soleus has been reanalyzed. Only darkly stained SDH fibers are treated as SDH positive fibers.

3rd Editorial Decision

26 June 2019

Thank you for submitting your revised manuscript. I have now looked at everything and all is fine. Therefore I am very pleased to accept your manuscript for publication in EMBO Reports.

Congratulations on the very nice work!

Corresponding Author Name:Gang Shu

Journal Submitted to: EMBOR

Manuscript Number:EMBOR-2019-47892